# *FAM111B* Acts as an Oncogene in Bladder Cancer

**DOI:** 10.3390/cancers15215122

**Published:** 2023-10-24

**Authors:** Ning Huang, Lei Peng, Jiaping Yang, Jinqian Li, Sheng Zhang, Mingjuan Sun

**Affiliations:** 1Department of Biochemistry and Molecular Biology, College of Basic Medical Sciences, Naval Medical University, Shanghai 200433, China; huangning.bio@foxmail.com (N.H.); penglei20010428@foxmail.com (L.P.); smmu_shenghuayjp@163.com (J.Y.); lijinqian1991@foxmail.com (J.L.); 2Medical Oncology, Shanghai Cancer Center, Fudan University, Shanghai 200032, China

**Keywords:** *FAM111B*, bladder cancer, proliferation, migration, apoptosis, oncogene

## Abstract

**Simple Summary:**

Bladder cancer can be categorized into non-muscle- and muscle-invasive bladder cancer based on the extent of invasion. While non-muscle-invasive bladder cancer is associated with a low mortality rate, it displays a high recurrence rate, and more than half of the patients are at risk of progressing to muscle-invasive bladder cancer. Traditional treatments, such as surgery and chemotherapy, significantly impact the patients’ quality of life. However, targeted therapies offer potential breakthroughs in the treatment of bladder cancer. Our study identified *FAM111B* as an oncogene that promotes the tumorigenesis, progression, and metastasis of bladder cancer. Thus, *FAM111B* gene is expected to serve as a promising molecular target for the therapy of bladder cancer.

**Abstract:**

Bladder cancer (BLCA) is a prevalent malignancy of the urinary system, associated with a high recurrence rate and poor prognosis. *FAM111B*, which encodes a protein containing a trypsin-like cysteine/serine peptidase domain, has been implicated in the progression of various human cancers; however, its involvement in BLCA remains unclear. In this study, we investigated the expression of *FAM111B* gene in tumor tissues compared to para-tumor tissues using immunohistochemistry and observed a significantly higher *FAM111B* gene expression in tumor tissues. Furthermore, analysis of clinical characteristics indicated that the increased *FAM111B* gene expression correlated with lymphatic metastasis and reduced overall survival. To investigate its functional role, we employed *FAM111B*-knockdown BLCA cell models and performed cell proliferation, wound-healing, transwell, and flow cytometry assays. The results showed that decreased *FAM111B* gene expression inhibited proliferation and migration but induced apoptosis in BLCA cells. In vivo experiments further validated that *FAM111B* knockdown suppressed tumor growth. Overall, our findings suggest that *FAM111B* acts as an oncogene in BLCA, playing a critical role in tumorigenesis, progression, and metastasis of BLCA. In conclusion, we have demonstrated a strong correlation between the expression of *FAM111B* gene and the development, progression, and metastasis of bladder cancer (BLCA). Thus, *FAM111B* is an oncogene associated with BLCA and holds promise as a molecular target for future treatment of this cancer.

## 1. Introduction

Bladder cancer (BLCA) ranks among the top ten most prevalent cancers globally, with about 550,000 new cases and nearly 220,000 related fatalities reported annually [1]. Risk factors such as older age, male gender, and history of smoking are associated with the development of BLCA [2]. Approximately 75% of patients diagnosed with BLCA exhibit non-muscle-invasive bladder cancer (NIMBC), while the remaining 25% have muscle-invasive bladder cancer (MTBC) [3]. Bacillus Calmette and Guérin (BCG) is the gold standard in treatment of intermediate and high-risk NMIBC [4]. Nevertheless, approximately 40% of patients with NMIBC do not respond to BCG treatment, and in certain cases, it even triggers side effects [5]. Radical cystectomy (RC) and cisplatin-based chemotherapy are the main protocols for patients with MIBC; however, their effectiveness is also unsatisfactory [6,7,8]. Therefore, novel clinical approaches are urgently required for BLCA treatment. Targeted therapies have achieved remarkable success in a wide range of cancers; however, their use in BLCA is still in the preliminary stages [9]. Thus, identifying new molecular targets may offer a potential solution for treating BLCA in the future.

Family with sequence similarity 111 member B (*FAM111B*) encodes a protein containing a trypsin-like cysteine/serine peptidase domain; however, its cellular functions remain unclear [10]. Hereditary fibrosing poikiloderma with tendon contractures, myopathy, and pulmonary fibrosis (POIKTMP) is the first reported disease linked to *FAM111B* gene [11]. In recent years, there is increasing evidence suggesting *FAM111B* gene is relevant to tumorigenesis and development, including in pancreatic cancer [12], lung adenocarcinoma (LUAD) [13,14], breast cancer [15], papillary thyroid cancer (PTC) [16], ovarian cancer [17], hepatocellular carcinoma (HCC), and gastric cancer [18,19]. Consequently, it may represent a promising therapeutic target for various cancers. Seo et al. discovered that *FAM111B* mutation can lead to inherited exocrine pancreatic dysfunction, increasing susceptibility to pancreatic cancer [12]. Sun et al. highlighted the essential function of *FAM111B* gene in the malignant progression of LUAD and proposed that p53 may be upstream of *FAM111B* gene [13]. Based on this, Li et al. found that FAM111B protein may in turn lead to p53 degradation, while knockdown of *FAM111B* probably suppresses the proliferation and migration of hepatoma cells by activating the p53 pathway [18]. *FAM111B* is generally a cancer-promoting gene in most tumor types, with the exception of its ability to suppress PTC. DNMT3B-mediated methylation decreases the expression of *FAM111B* gene, which consequently promotes tumor growth and metastasis [16]. In addition, *FAM111B* gene has been proven to be a reliable signature for predicting HCC response to transarterial chemoembolization (TACE) [20], and it can also be used to anticipate distal metastasis in patients with cervical cancer [21]. Nonetheless, the impact of *FAM111B* gene in BLCA remains unknown.

Therefore, we aimed to assess the role of *FAM111B* gene in bladder carcinogenesis and progression. Initially, we performed an immunohistochemical analysis to compare the difference in *FAM111B* gene expression levels between tumor and para-tumor tissues. Subsequently, we evaluated the impact of *FAM111B* gene expression on the survival risk for patients with BLCA. We then generated *FAM111B*-knockdown BLCA cell models in vitro to examine the effects of reduced *FAM111B* gene expression on the proliferation, migration, and apoptosis of BLCA cells. We eventually validated the role of *FAM111B* gene in tumor growth in mice by xenograft experiments.

## 2. Materials and Methods

### 2.1. Sample Collection and Immunohistochemical Analysis

From Shanghai Outdo Biotechnology Co., Ltd. (Shanghai, China). It included 46 cases of BLCA tissues and 10 pairs of BLCA tissues and matched adjacent tissues. First, deparaffinization of the tissue microarray was carried out, followed by antigen retrieval with 1× citrate acid buffer and blocking of endogenous peroxidase with 3% H_2_O_2_. Next, rabbit anti-FAM111B (1:100, Thermo, Waltham, MA, USA, Cat. PA5-28529) was added and incubated at 4 °C overnight. After elution with 1× PBS, goat anti-rabbit IgG H & L (HRP) (1:400, Abcam, Waltham, MA, USA, Cat. ab97080) was added and incubated at 4 °C overnight. The tissue sections were then stained with diaminobenzidine (DAB) for 5 min and re-stained with hematoxylin for 10–15 s. Finally, a total of 64 tissue sections were successfully stained and the images were captured and analyzed by light microscope (Nexcope, Ningbo, China, NE610). The positive cell score was graded as 0 (0%), 1 (1–24%), 2 (25–49%), 3 (50–74%), 4 (75–100%), and the staining intensity score was graded as 0 (no signal color), 1 (light yellow), 2 (brownish yellow), 3 (dark brown). The IHC score is calculated by multiplying the positive cell score by the staining intensity score. The higher the IHC score, the higher the *FAM111B* gene expression level.

### 2.2. Cell Lines and Cell Culture

We obtained a normal bladder epithelium cell line, HCV-29 (Cat. ZY6561), from Shanghai Zeye Biotechnology Co., Ltd. (Shanghai, China). The cells were cultured in Roswell Park Memorial Institute (RPMI) 1640 complete medium containing 10% fetal bovine serum (FBS) (Ausbian, Thornton, NSW, Australia, Cat. A11-102). Additionally, we sourced four urinary BLCA cell lines, including EJ (Cat. iCell-h059), 5637 (Cat. iCell-h232), RT4 (Cat. iCell-h184), T24 (Cat. iCell-h208), from iCell Bioscience Inc. (Shanghai, China). EJ, 5637 and T24 cells were cultured in RPMI 1640 complete medium supplemented with 10% FBS, while RT4 cells were cultured in McCoy’s 5A medium with 10% FBS. A stable temperature (37 °C) and CO_2_ level (5%) were provided for cell growth.

### 2.3. Construction of Lentiviral shRNA Vectors and Transfection

Three short hairpin RNAs (shRNAs) were designed based on the *FAM111B* gene. Linear vectors BR-V108 (Shanghai YiBeiRui Biomedical Science and Technology Co., Ltd., Shanghai, China) were prepared by restriction enzymes AgeI (NEB, Ipswich, MA, USA, Cat. R3552L) and EcoRI (NEB, USA, Cat. R3101L). These two sequences were ligated via T4 DNA ligase, and the recombinant vectors were transformed into TOP10 *Escherichia coli* for amplification. Positive clones were selected using PCR. After the construction of lentiviral shRNA vectors, the lentiviral vectors, packaging plasmids (psPAX2), and envelope plasmid (pMD2.G) were co-transfected into 293T cells and cultured for 48 h [22,23]. Lentiviruses were harvested from the cells and subjected to purification and concentration. Subsequently, they were used to transfect T24 and EJ cell lines. Both qPCR and Western blot were utilized to measure *FAM111B* knockdown efficiency. Only the most effective lentivirus (target sequence: AGCAAAGAAGATGGACACATA) was used for subsequent experiments.

### 2.4. Real-Time Quantitative PCR (qPCR)

We extracted total RNA by Trizol reagent (Sigma-Aldrich, St. Louis, MO, USA, Cat. T9424) following the manufacturer’s instructions, and utilized a NanoDrop 2000/2000c spectrophotometer (Thermo, USA) to determine the quantity and quality of RNA. Hiscript Q RT supermix for qPCR (+gDNA WIPER) (Vazyme, Nanjing, China, Cat. R123-01) was used to synthesize cDNA by reverse transcription, and AceQ qPCR SYBR Green Master Mix (Vazyme, Nanjing, China, Cat. Q111-02) was used for cDNA amplification. Finally, the 2-ΔΔCt method was selected for data analysis, and GAPDH was selected as an internal reference. The primer sequences used in this study were as follows: *FAM111B*, CCAGACAATTCCCAGGATTAGA (forward), TAGCATACCGCCTACCCAGA (reverse); *GAPDH*, TGACTTCAACAGCGACACCCA (forward), CACCCTGTTGCTGTAGCCAAA (reverse).

### 2.5. Western Blot (WB)

First, the cells were lysed by radio immunoprecipitation assay (RIPA) lysis buffer and the total protein was isolated; the protein concentration was measured using the Pierce BCA Protein Assay Kit (Thermo, USA, Cat. 23225). The proteins were then separated by SDS-PAGE and transferred to polyvinylidene fluoride (PVDF) membranes at a constant current of 300 mA for 45 min at 4 °C. TBS + Tween (TBST) solution containing 5% skim milk was used to block the PVDF membranes; the primary antibody was added individually and incubated overnight at 4 °C. After washing with TBST solution, the PVDF membranes were then incubated with secondary antibody for 1 h at room temperature. Finally, the blots were visualized by immobilon Western Chemiluminescence HRP Substrate Kit (Millipore, Burlington, MA, USA, Cat. WBKLS0050) and the chemiluminescence was evaluated by chemiluminescence imager (GE, Boston, MA, USA, AI600). The primary antibodies used in WB were rabbit anti-FAM111B (1:1000, Thermo, USA, Cat. PA5-28529) and mouse anti-GAPDH (1:30,000, Proteintech, Rosemont, IL, USA, Cat. 60004-1-lg). The secondary antibodies used for WB were goat anti-rabbit IgG (H + L) (1:3000, Beyotime, Shanghai, China, A0208) and goat anti-mouse IgG (H + L) (1:3000, Beyotime, Haimen, China, A0216).

### 2.6. Celigo Cell Counting Assay

Lentivirus-transfected T24 and EJ cells were digested with trypsin and resuspended in complete medium. The cells were then seeded onto 96-well culture plates at a density of 2 × 10^3^ cells/well and cultured in an incubator at 37 °C, 5% CO_2_. The experimental group and negative control each contained three replicate wells. The Celigo image cytometer (Nexcelom Bioscience, Lawrence, MA, USA) was used to detect green fluorescent protein (GFP) signal for 5 consecutive days, and ImageJ was used to count cells by processing and analyzing images. We plotted the cell proliferation curve using the average cell count of three replicate wells. The fold change of cell proliferation was calculated by dividing the fold cell count of the negative control on the last day by that of the experimental group.

### 2.7. Wound-Healing Assay

The T24 and EJ cells transfected with lentivirus were seeded onto 96-well culture plates at a density of 5 × 10^4^ cells/well and cultivated at 37 °C, 5% CO_2_ in a cell culture incubator. The experimental group and negative control contained three replicate wells, respectively. On the next day, the medium was replaced with RPMI 1640 containing 0.5% FBS. A scraper was used to align the lower central part of the well and was gently pushed upward to form scratches. After 2–3 washes with serum-free medium, medium with 0.5% FBS was added. Cellomics (Thermo, USA, ArrayScan VT1) was used to record the cell position at specific hours (T24: 0 h, 4 h, 8 h; EJ: 0 h, 8 h, 16 h) and calculate the migration rate of the cells.

### 2.8. Transwell Assay

Corning transwell chambers (Cat. 3422) were used to perform the transwell migration assay. Three replicate wells were used for each experimental group and negative control. First, 100 μL serum-free medium was added to the upper chamber, left for 1–2 h, and then removed. Then, 600 μL medium containing 30% FBS was added to the bottom chambers. The T24 and EJ cells transfected with lentivirus were digested with trypsin and resuspended before being transferred to the upper chambers (5 × 10^4^ cells/well). After 24 h of incubation, the medium was discarded, and non-migratory cells were wiped off with cotton swabs. Migratory cells on the lower surface of the polycarbonate (PC) membrane were stained with crystal violet for 5 min. The cells were observed under the microscope, and the cells in each field were counted using ImageJ 1.8.0.

### 2.9. Cell Apoptosis Assay

The relationship between *FAM111B* gene expression level and cell apoptosis was investigated by detecting cell state after *FAM111B* knockdown. Lentivirus-transfected T24 and EJ cells were digested, resuspended, and centrifuged at 1300 rpm for 5 min. The cell precipitate was washed with 4 °C pre-cooled D-Hanks (pH = 7.2–7.4) and 1× binding buffer. The cell suspension was then centrifuged at 1300 rpm for 3 min to collect the cells. After resuspension with 1 × binding buffer, 10 μL Annexin V-APC was added and kept in the dark at room temperature for 10–15 min. Finally, a flow cytometer (Millipore, USA, Guava easyCyte H) was utilized to identify apoptotic cells. Both the experimental group and the negative control contained three replicated wells.

A human apoptosis antibody array (Abcam, USA, Cat. Ab134001) was used to detect the expression levels of 43 human apoptosis markers [24,25,26]. T24 cells transfected with lentivirus were washed with PBS and lysed using cell lysis buffer. Total protein was then extracted, and the blocked antibody membrane was incubated with protein sample overnight at 4 °C. Next, 1 × biotin-conjugated anti-cytokines were added and incubated overnight at 4 °C. This was followed by 1 × streptavidin-HRP addition and incubation at room temperature for 2 h. Finally, the protein signals were detected by chemiluminescence imaging system and Image J was used to obtain spot signal densities.

### 2.10. Construction of a Cell-Line-Derived Xenograft Model

A total of 20 BALB/c female nude mice (4 weeks old) were purchased from Beijing Vital River Laboratory Animal Technology Co., Ltd. (Beijing, China) and randomly divided into the experimental group and negative control. *FAM111B* knockdown T24 cells and control cells were injected subcutaneously into the mice (4 × 10^6^ cells/mouse). Starting on day 10 after injection, tumor volumes were recorded every 5 days. On the 30th day, 0.7% pentobarbital sodium was injected intraperitoneally at a dose of 10 μL/g to anesthetize the mouse. An in vivo imaging system (Berthold, Bad Wildbad, Germany, LB983) was used to observe fluorescence and measure tumor burden. All 20 mice were then sacrificed, the tumors were removed, and the volume and weight of each tumor tissue were recorded.

### 2.11. Ki-67 Staining

Ki-67 is a nuclear protein related to the cell cycle, and its high expression indicates strong proliferative activity of tumor cells. The tumor tissues from mice were first sectioned and pre-processed for antigen retrieval and blocking. Next, the primary antibody anti-Ki67 (1:300, Abcam, USA, Cat. ab16667) was added and incubated overnight at 4 °C, the secondary antibody goat anti-rabbit IgG H & L (HRP) (1:400, Abcam, USA, Cat. ab97080) was added and incubated for 2 h at room temperature in the dark. Finally, the slides were stained with hematoxylin and eosin and observed under the microscope.

### 2.12. Statistical Analysis

R 4.3.1 and GraphPad Prism 8.2.1 were used for statistical analyses. Kaplan–Meier survival analysis was performed using the survival and survminer package. Experimental data are presented as mean ± SD. Statistical differences were measured by *t*-test, *p* < 0.05 was considered significant. All assays were performed in three independent experiments.

## 3. Results

### 3.1. The Expression of FAM111B Gene Was Positively Correlated to BLCA Progression

To investigate the expression pattern of *FAM111B* gene in BLCA tissues, we obtained a total of 66 samples (56 tumor tissues and 10 para-tumor tissues) from 56 patients with BLCA. We then used immunohistochemistry (IHC) to detect the expression levels of *FAM111B* gene. Of them, two tumor tissue samples were invalid due to section detachment, and the remaining 64 samples were used for subsequent analysis. As depicted in Figure 1A, the FAM111B protein is mainly located in the nucleus. The IHC scores in tumor tissues were generally greater than in para-tumor tissues (*p* < 0.001, Figure 1A,B). Moreover, tumor tissues exhibited higher IHC scores than their paired adjacent tissues (*p* < 0.001, Figure 1C), indicating the upregulation of *FAM111B* gene expression in tumor tissues.

Next, we divided the patients into two groups based on median IHC score and analyzed the relationship between *FAM111B* gene expression and clinical characteristics. Patients exhibiting high levels of *FAM111B* gene expression were found to be more prone to suffering from lymphatic metastasis (Figure 1A). This was supported by Fisher’s exact test (*p* < 0.05, Table 1) and confirmed by the Spearman correlation test, which demonstrated a positive link between *FAM111B* gene expression and lymphatic metastasis (r = 0.342, *p* < 0.05). Furthermore, our Kaplan–Meier survival analysis highlighted a significant divergence in overall survival (OS) between the two groups (*p* < 0.01, Figure 1D). Hyper-expression of *FAM111B* gene was associated with a shorter OS. Therefore, upregulation of *FAM111B* gene exprssion may contribute to the progression of BLCA. These findings suggest that *FAM111B* gene plays a vital role in the occurrence and development of BLCA.

### 3.2. Construction of a FAM111B-Knockdown BLCA Cell Model

We constructed a *FAM111B*-knockdown BLCA cell model for follow-up studies to assess the specific biological function of *FAM111B* gene in BLCA progression. Initially, we measured the mRNA levels of *FAM111B* in five cell lines, comprising a non-malignant bladder epithelium HCV-29 and four urinary BLCA cell lines (EJ, 5637, RT4, T24), using qPCR analysis. As anticipated, *FAM111B* gene was found to be expressed significantly higher in BLCA cell lines than in regular bladder epithelia (Figure 2A). Among the BLCA cell lines, T24 and EJ showed the highest expression of *FAM111B* gene. Then, three shRNAs were designed to knock down *FAM111B* by lentiviral transfection. shFAM11B-1 was the most effective shRNA in T24 and was thus selected for further experiment (Figure 2B). mRNA levels of *FAM111B* in T24 and EJ decreased by 55.63% (*p* < 0.05) and 66.89% (*p* < 0.001), respectively, after transfection with shFAM111B-harboring lentivirus compared with the negative control (shCtrl) (Figure 2C,D). Consistently, *FAM111B* gene in T24 and EJ were significantly downregulated (Figure 2C,D and Appendix A). These findings provide evidence of successful construction of a BLCA cell model with *FAM111B* knockdown.

### 3.3. Knockdown of FAM111B Inhibited Cell Proliferation, Migration In Vitro

To elucidate the effect of *FAM111B* knockdown on BLCA cell lines, we counted the cell numbers of shFAM111B and shCtrl groups for 5 days by detecting the GFP signal and found that cell proliferation of T24 and EJ was significantly attenuated after *FAM111B* knockdown (T24: fold change = −2.3, *p* < 0.001, Figure 3A; EJ: fold change = −4.8, *p* < 0.001, Figure 3B). The result showed that *FAM111B* gene functioned as a promoter of BLCA cell proliferation.

In addition, we measured the capacity of cell migration by wound-healing. For T24 cells, the shFAM111B group showed a 50% decrease in migration rate at both 4 and 8 h (4 h: *p* < 0.01; 8 h: *p* < 0.001, Figure 3C). For EJ cells, the migration rate of the shFAM111B group decreased by 12% (*p* < 0.05) and 24% (*p* < 0.01) at 8 and 16 h, respectively (Figure 3D). Obviously, knockdown of *FAM111B* could impair the migration ability of the BLCA cell lines. The transwell migration assay confirmed this result. The number of migratory cells was significantly reduced in both cell lines after *FAM111B* knockdown (*p* < 0.001, Figure 3E,F).

### 3.4. Knockdown of FAM111B Induced Cell Apoptosis In Vitro

Flow cytometry was performed to identify the apoptotic cells to further explore the effect of *FAM111B* knockdown on BLCA cell lines. For both T24 and EJ cells, the proportion of apoptotic cells dramatically increased in the shFAM111B group compared with the shCtrl group (T24: fold change = 8.3, *p <* 0.001; EJ: fold change = 16.2, *p <* 0.001, Figure 4A,B). The result suggested that *FAM111B* gene might be involved in regulation of cell apoptosis.

Moreover, we analyzed the protein expression levels of 43 human apoptotic markers in T24 cells using a human apoptosis antibody array. Among these proteins, eight were found to have significantly elevated expression levels after *FAM111B* knockdown, including BID, BIM, Caspase8, CD40, CD40L, cytoC, DR6, IGFBP6 (*p* < 0.05 and FC > 1.2, Figure 4C). Notably, Caspase8 and CD40 were the most upregulated, which might be the direct cause of cell apoptosis. The result indicated that *FAM111B* knockdown could induce cell apoptosis by upregulating the expression of apoptosis-related proteins in vitro.

### 3.5. Knockdown of FAM111B Suppressed Tumor Growth In Vivo

To evaluate the consistency of *FAM111B* knockdown effects in vivo and in vitro, we established xenograft models by implanting T24 cells transfected with shFAM111B or shCtrl into immunocompromised mice. We monitored the tumor volume every five days from the tenth day after injection and observed that the average tumor volume of xenograft mice injected with *FAM111B* knockdown cells increased much more slowly than those injected with control cells (Figure 5A). At thirty days post-injection, the result of in vivo bioluminescence imaging suggested that only half of the xenograft mice (5/10) in the shFAM111B group developed macroscopic tumors, while tumors were observed in all xenograft mice (10/10) in the shCtrl group. Total fluorescence was higher in the shCtrl group than in the shFAM111B group, indicating larger tumors in the shCtrl group (*p* < 0.05, Figure 5B and Appendix A). Tumors in the shFAM111B group were also lighter in weight (*p* < 0.01, Figure 5C). These results suggested that downregulation of *FAM111B* gene expression could suppress tumor growth.

We further conducted immunohistochemical staining for Ki-67 using tumor tissue. The shFAM111B group exhibited a significant decrease in the expression of Ki-67, suggesting that *FAM111B* knockdown weakened the proliferative capacity of BLCA cells in vivo (Figure 5D).

## 4. Discussion

BLCA is the most common type of cancer affecting the urinary system. On the basis of the degree of invasion, BLCA can be classified into NMIBC and MIBC. NMIBC has a low mortality rate but is prone to recurrence and up to 53% cases of NMIBC are at risk of progression to MIBC [27]. MIBCs have a 30% lower five-year survival rate than NMIBC and is more likely to metastasize [28]. Lymphatic metastasis is the predominant type of metastasis in BLCA, leading to an extremely poor prognosis [9]. Therefore, identifying molecular targets that specifically target lymphatic metastasis could be essential for reducing mortality rates in BLCA [9]. In fact, several genes and proteins have demonstrated a correlation with BLCA prognosis. For example, NMP22 is a nuclear matrix protein which has Food and Drug Administration (FDA) approval for detection and surveillance of BLCA [29]. Furthermore, genes including *ENO1* [30], *NXPH4* [31], and *FADS1* [32] could potentially serve as prognostic biomarkers for BLCA. Nevertheless, the molecular targets that can effectively treat BLCA are not yet established.

In the present study, we identified *FAM111B* as an oncogene for BLCA. The *FAM111B* gene expression was significantly elevated in tumor tissues, which was consistent with the results observed in The Cancer Genome Atlas (TCGA) [19]. Moreover, there was a significant positive correlation between lymphatic metastasis and *FAM111B* gene expression level, indicating that *FAM111B* gene may play a role in promoting the metastatic process of BLCA. In the future, *FAM111B* gene has the potential to monitor BLCA progression. We also observed that patients with BLCA with high *FAM111B* gene expression tended to have shorter OS. However, this contradicts the pan-cancer analysis, which suggests that *FAM111B* gene has no significant impact on the overall survival of BLCA patients [19]. This incongruity is probably because our study used the immunohistochemical score to measure the expression of *FAM111B* gene, whereas the earlier study employed the quantity of transcribed mRNA. Comparatively, immunohistochemistry can directly reflect the protein expression of *FAM111B*.

Inhibition of *FAM111B* gene expression results in decreased BLCA cell proliferation and tumor growth. This finding is supported by positive correlation between *FAM111B* gene expression and the cell proliferation marker, Ki-67. The process of epithelial-mesenchymal transition (EMT) has been widely demonstrated to facilitate lymphatic metastasis [9,27]. Suppression of *FAM111B* has been shown to effectively modify the expression of EMT-related proteins, including E-Cadherin upregulation and N-Cadherin and Vimentin downregulation, to inhibit EMT [15]. This is consistent with our finding that *FAM111B* gene could promote the migration of BLCA cells. Therefore, we hypothesize that *FAM111B* gene may aid in the progression of BLCA lymphatic metastasis through the EMT process. However, the connection between *FAM111B* gene expression and lymphatic metastasis and its underlying mechanism requires additional research for confirmation.

Additionally, we found that suppression of *FAM111B* significantly induced apoptosis in BLCA cells by increasing apoptosis-promoting proteins. BID and BIM are proapoptotic proteins of the Bcl-2 family that can increase permeability of the mitochondrial membrane and stimulate the mitochondrion to release proapoptotic factors such as cytoC [28]. The full-length BID is located in the cytosol, whereas the truncated BID (tBID) moves to the mitochondria, enabling it to transmit apoptotic signals from the cytoplasmic membrane to the mitochondria [33]. BIM could also act upstream of SLC7A11 and GPX4 to mediate abivertinib-induced ferroptosis [34]. Caspase-8 is an initiator caspase, and its elevated level indicates the initiation of apoptosis [35]. Fritsch et al. have recently discovered that Caspase-8 is actually the molecular switch for apoptosis, necroptosis, and pyroptosis [36]. CD40 and DR6 are both members of the tumor necrosis factor receptor superfamily (TNFRS). The interaction between CD40 and CD40L (CD154) is critical to the workings of the immune system [37], and CD40-mediated apoptosis has been demonstrated in multiple research papers [38,39,40]. Furthermore, IGFBP-6 has been demonstrated to have a significant role in cell migration and apoptosis [41]. It is evident that these apoptosis-promoting proteins may result in other types of cell death, and in subsequent research, we will investigate the affiliation between *FAM111B* gene expression and other types of cell death.

The major limitation of our study is that we did not further investigate the pathways involved in *FAM111B* gene that facilitate BLCA progression and the related molecular mechanisms. Indeed, numerous studies exist exploring the mechanism of action of *FAM111B* gene in other tumors. *FAM111B* gene has been identified as a direct target of p53; its protein can also degrade p53 and inhibit the p53 pathway [13,18]. Kawasaki et al. discovered that FAM111B protein can degrade p16, thus prolonging G0/G1 arrest [14]. Li et al. demonstrated the transcription factor YY1’s capacity to bind to the *FAM111B* promoter, enhancing *FAM111B* gene expression and facilitating the development of breast cancer [15]. However, these studies are not flawless, and the impact of *FAM111B* gene on tumors still requires further clarification.

## 5. Conclusions

In summary, we have outlined the robust correlation between *FAM111B* gene expression and tumorigenesis, progression, and metastasis of BLCA. *FAM111B* is an oncogene for BLCA and may be a prospective molecular target for future treatment of BLCA.

## Figures and Tables

**Figure 1 cancers-15-05122-f001:**
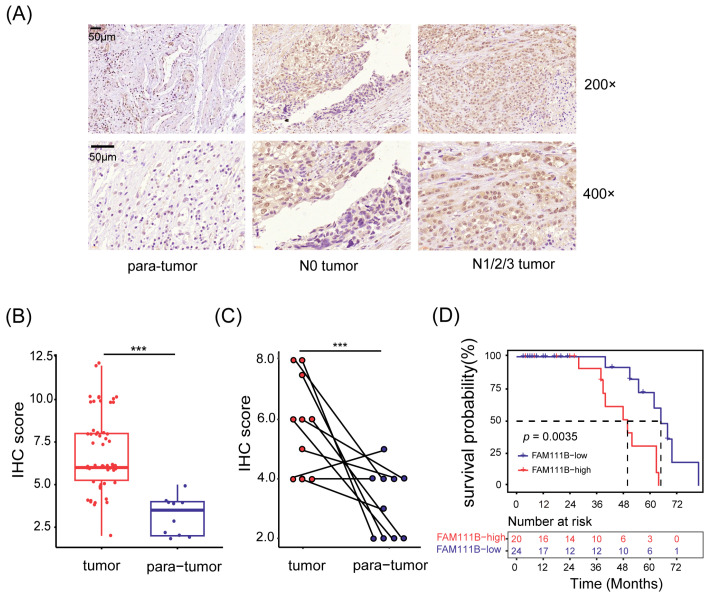
*FAM111B* gene was highly expressed in BLCA tumor tissues. (**A**) Representative immunohistochemical images of tissue sections. N0 tumor: BLCA without lymphatic metastasis; N1/2/3 tumor: BLCA with lymphatic metastasis. Magnification: 200× and 400×. (**B**) Comparison of the IHC scores between tumor and para-tumor tissues. (**C**) The IHC scores of tumor tissues and their paired adjacent tissues. (**D**) Kaplan–Meier survival analysis of OS. The dotted line shows median survival time. *** *p* < 0.001.

**Figure 2 cancers-15-05122-f002:**
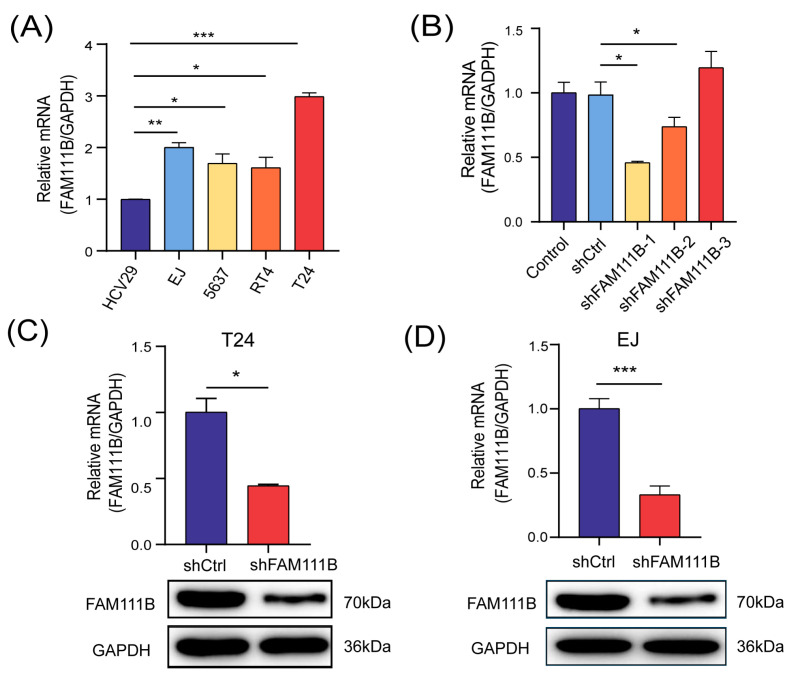
*FAM111B*-knockdown BLCA cell model was successfully constructed. (**A**) The mRNA levels of *FAM111B* among HCV-29 and four urinary BLCA cell lines (EJ, 5637, RT4, T24). (**B**) The knockdown efficiency of three shRNAs. The mRNA and protein levels of *FAM111B* in T24 (**C**) and EJ (**D**) after transfecting with shFAM111B-harboring lentivirus. * *p* < 0.05; ** *p* < 0.01; *** *p* < 0.001.

**Figure 3 cancers-15-05122-f003:**
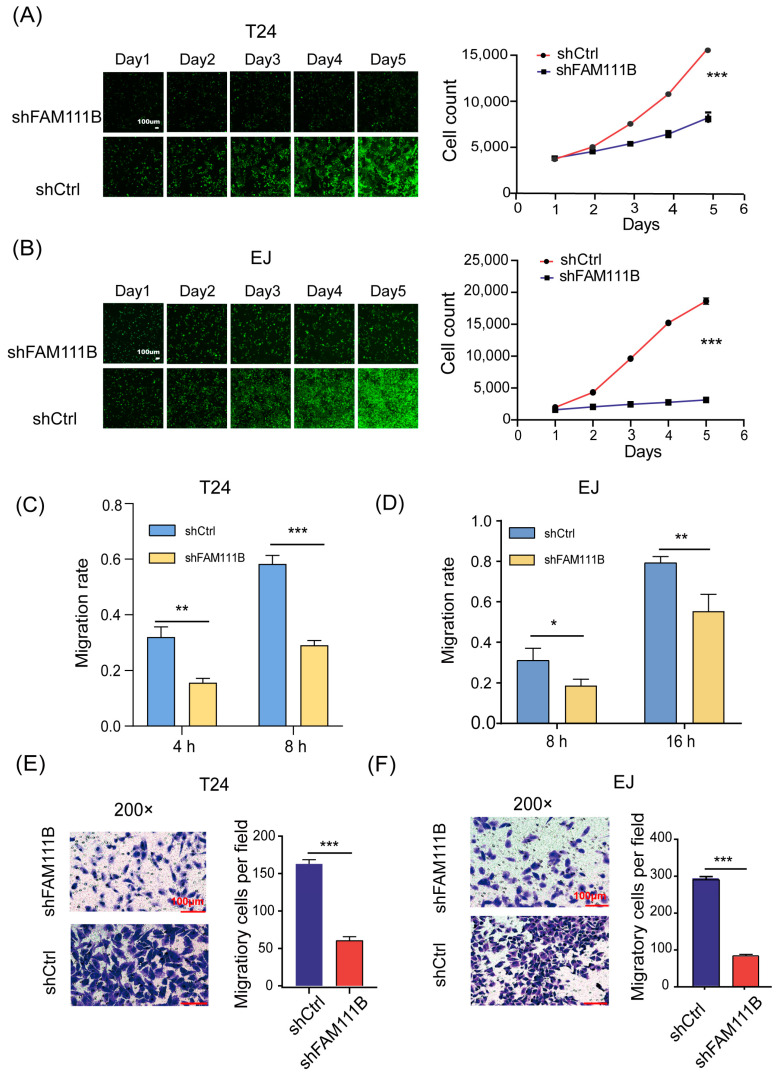
Knockdown of *FAM111B* inhibited cell proliferation and migration. The proliferation rates of shFAM111B and shCtrl groups in T24 (**A**) and EJ (**B**) cells. The capacity of cell migration in T24 (**C**) and EJ (**D**) detected by wounding-healing. The ability of cell migration in T24 (**E**) and EJ (**F**) measured by transwell migration assay. Magnification: 200×. * *p* < 0.05; ** *p* < 0.01; *** *p* < 0.001.

**Figure 4 cancers-15-05122-f004:**
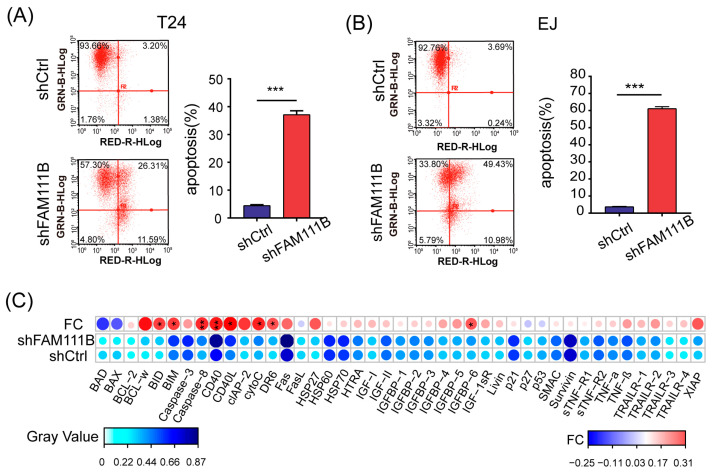
Knockdown of *FAM111B* induced cell apoptosis in vitro. The proportion of apoptotic cells in T24 (**A**) and EJ (**B**) detected by flow cytometry. GRN-B-HLog: Green-B Fluorescence; RED-R-HLog: Red-R Fluorescence. (**C**) The protein levels of 43 human apoptotic markers in shFAM111B and shCtrl groups in T24. Fold change (FC) = shFAM111B group/shCtrl group. * *p* < 0.05; ** *p* < 0.01; *** *p* < 0.001.

**Figure 5 cancers-15-05122-f005:**
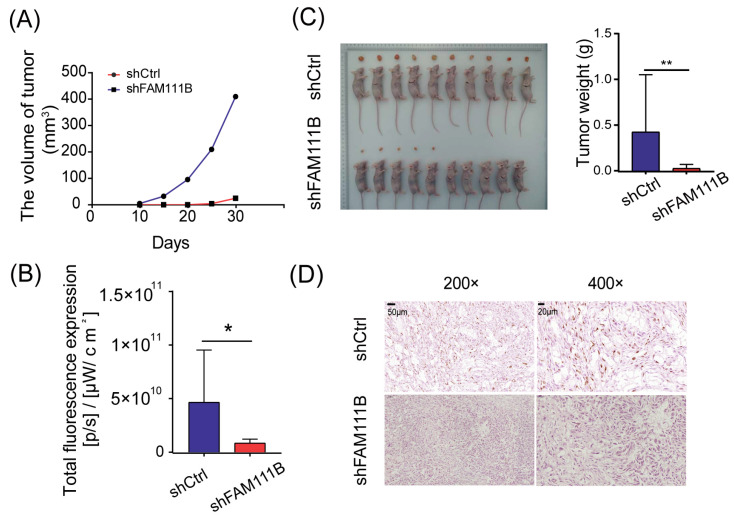
Knockdown of *FAM111B* suppressed tumor growth in vivo. (**A**) The average tumor volumes of xenograft mice injected with *FAM111B* knockdown cells and control cells. (**B**) Total fluorescence expression of experimental and control groups (**C**) The tumor weights of xenograft mice injected with *FAM111B* knockdown cells and control cells. (**D**) The expression of Ki-67 in experimental and control groups detected by immunohistochemical staining. Magnification: 200× and 400×. * *p* < 0.05; ** *p* < 0.01.

**Table 1 cancers-15-05122-t001:** Correlation between *FAM111B* gene expression levels and clinical characteristic of patients with BLCA.

Characteristic	No. of Patients	Low *FAM111B*	High *FAM111B*	*p*
All patients	54	30	24	
Age (years)				0.085
<72	27	18	9	
≥72	27	12	15	
Gender				0.210
Male	46	24	22	
Female	8	6	2	
Tumor size				0.054
<4 cm	22	16	6	
≥4 cm	30	14	16	
Grade				0.359
II	16	10	6	
III	38	20	18	
AJCC stage				0.079
I/II/III	30	20	10	
IV	7	2	5	
Tumor infiltrate (T)				0.143
T1/T2	24	12	12	
T3/T4	23	16	7	
Lymphatic metastasis (N)				0.045 *
N0	33	21	12	
N1/2/3	6	1	5	

** p* < 0.05.

## Data Availability

All data in this study can be obtained from the corresponding author upon reasonable request.

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
