# Peer review of "FAM111B Acts as an Oncogene in Bladder Cancer"

_cancers, 2023, doi:10.3390/cancers15215122_

Round 1

Reviewer 1 Report

The study by Huang et al. reveals for the first time the role of FAM111B in bladder cancer and may provide a new target for targeted therapy of bladder cancer. The article is innovative to a certain extent and can arouse readers' interest, but there are still some problems in the manuscript itself that may make readers suspicious.

1.        The introductions of apoptosis-related molecules such as caspase need more literature to support.

2.        Plotting scales are needed to add in Figure2 E/F.

3.        The magnification of Figure3 C/D should be reflected in both the figure and the note.

4.        Both the gene names and protein names need to be standardized.

5.        Most of the immunohistochemical results in Figure1 showed connective tissue rather than cancer tissue, suggesting that appropriate visual field replacement should be selected

6.        Figure3 is titled "Knockdown of FAM111B inhibited cell proliferation, migration in vitro", but the resulting figure only shows transwell, scratches and other migration-related results, which does not reflect the influence on proliferation. It is recommended to supplement proliferation-related experiments, such as cck8, clone formation, Ki67 detection, etc.

7.        The FC results in Figure4 are not clearly marked, so the quality and clarity of the pictures should be improved

It is suggested that the author carefully review the content of the article, or invite colleagues to polish the language of the article to avoid grammatical errors. 

Author Response

Dear Reviewer:

Thank you very much for your professional review of our article. As you pointed out, there are several problems that need to be addressed. Following your kind suggestions, we have made extensive corrections to our previous draft, including replacing problematic images and adding references. Meanwhile, the manuscript had be reviewed and edited by language services of ELSEVIER.

Your comments are laid out below in italicized font and our response is given in normal font. The changes/additions to the manuscript are given in blue text.

Q1: The introductions of apoptosis-related molecules such as caspase need more literature to support.

The Human Apoptosis Antibody array (ab134001) is a commercially available array that has been utilized in numerous studies related to apoptosis. The relevant references have been added to the methods section (P4, Line 188). In addition, we have further elaborated on the functions of the apoptosis-related proteins in the discussion section and added supporting references (P12, Line376-390).

 Q2: Plotting scales are needed to add in Figure2 E/F.

The scale has been added to the images. The new images have been integrated into Figure 3 and labeled as Figure3 A/B.

Q3: The magnification of Figure3 C/D should be reflected in both the figure and the note.

The magnification has been reflected in both the figure and the note (P9, Line292). The new images have been labeled as Figure3 E/F.

Q4: Both the gene names and protein names need to be standardized.

We have uniformly set gene abbreviations in italics (e.g., FAM111B) and protein abbreviations in non-italics (e.g., FAM111B).

Q5: Most of the immunohistochemical results in Figure1 showed connective tissue rather than cancer tissue, suggesting that appropriate visual field replacement should be selected

We have selected new visual fields to replace the original Figure1 A

Q6: Figure3 is titled "Knockdown of FAM111B inhibited cell proliferation, migration in vitro", but the resulting figure only shows transwell, scratches and other migration-related results, which does not reflect the influence on proliferation. It is recommended to supplement proliferation-related experiments, such as cck8, clone formation, Ki67 detection, etc.

We have performed Celigo cell counting assay, which could be used to detect the capacity of cell proliferation. The result of assay was originally included in Figure 2 and is now included in Figure 3, labelled as Figure 3 A/B.

Q7: The FC results in Figure4 are not clearly marked, so the quality and clarity of the pictures should be improved.

We have improved the quality and clarity of flow cytometry images in Figure4.

Yours sincerely,

Ning Huang

October 16, 2023

Reviewer 2 Report

This is an interesting, thorough and well written manuscript presenting novel data on the role of FAM111B in bladder cancer and its potential as therapeutic target.

There are no major methodological issues and the presentation and interpretation of results is sound. I have only some minor comments on a few points that need clarification:

Line 91 the authors state ‘staining intensity was measured according to the IHC score’ – could they please give more details on the way the stain intensity was scored

2.   Line 141 two anti-FAM111B antibodies are listed – which one was used, and if two were used what was the reason for this?

3.    Paragraph 2.7 and 2.8. The wound healing and transwell assays were performed in replicate wells. But how many times were the experiment repeated?

4.    Line 222 the authors state ‘two tumour tissues were invalid’ – in which way? More details needed on why these samples were excluded from the analysis

5.    Figure 1 C: I find this graph confusing as it is meant to represent ten pairs of tumour and para-tumour tissue, however each data point is connected to several others and I cannot see 10 obvious pairs. This graph needs improving.

6.    Figure 1 D. I assume this is based on data on survival available for the patients whose tissue was used in the commercial tissue arrays. Please could you clarify if that’s the case.

Author Response

Dear Reviewer:

Thank you very much for your comments and professional advice. These opinions help to improve academic rigor of our article. Based on your suggestion, we have made corrected modifications on the revised manuscript.

Your comments are laid out below in italicized font and our response is given in normal font. The changes/additions to the manuscript are given in red text.

Q1: Line 91 the authors state ‘staining intensity was measured according to the IHC score’ – could they please give more details on the way the stain intensity was scored.

I apologize for the inaccuracy of this statement. The IHC scores are used to measure the expression of FAM111B. The higher the IHC score, the higher the FAM111B expression level. The IHC scores are calculated by multiplying the positive cell score by the staining intensity score. The scoring criteria for positive cell score and staining intensity score are described in detail in P2, Line88-92.

Q2: Line 141 two anti-FAM111B antibodies are listed – which one was used, and if two were used what was the reason for this?

We previously used different FAM111B antibodies in the T24 and EJ experiments. To standardize the results, we repeated the experiments using rabbit anti-FAM111B (1:1000, Thermo, USA, Cat. PA5-28529) and replaced the previous results (Figure2 C/D, Figure S1). The methods section was adjusted accordingly (P3, Line141-P4, Line 145).

Q3: Paragraph 2.7 and 2.8. The wound healing and transwell assays were performed in replicate wells. But how many times were the experiment repeated?

Three times. And the three independent experiments produced consistent results. We have added this information to the methods section (P5, Line 218).

Q4: Line 222 the authors state ‘two tumor tissues were invalid’ – in which way? More details needed on why these samples were excluded from the analysis

The tumor tissues were invalid due to section detachment. We have added this information to the results section (P5, Line224-225). Section detachment is common for IHC experiments. Possible reasons: 1. Excessive heat for antigen repair; 2. Excessive rinsing during the operation; 3. Problems with tissue microarray itself. We purchased several microarrays of the same model, they all appeared section detachment on the same position.

Q5: Figure 1 C: I find this graph confusing as it is meant to represent ten pairs of tumor and para-tumor tissue, however each data point is connected to several others and I cannot see 10 obvious pairs. This graph needs improving.

This is because identical IHC scores are present for various tissue sections. We have made modifications to Figure 1C

Q6: Figure 1 D. I assume this is based on data on survival available for the patients whose tissue was used in the commercial tissue arrays. Please could you clarify if that’s the case.

Yes. The commercial tissue microarray gives a complete view of the patients' pathological conditions, and we used the survival information to perform survival analyses. We have added the product number of microarray to the method section (P2, Line78).

Yours sincerely,

Ning Huang

October 16, 2023

Reviewer 3 Report

The manuscript describes the role of the potential oncogene in development of metastatic bladder cancer. This is important issue, especially because FAM111B was shown to play the role in carcinogenesis for other cancers. 

Immunohistochemical evaluation of tissue array is adequate and, indeed, shows the increase expression of FAM111B in metastatic samples. 

In vitro part of the study, to my mind must be substantially improved. 

1. Cell proliferation and apoptosis assay are important but the relation to other types of cell death will be helpful.

2. The results of cell movement assay are convincing, but since the relation to metastasis formation is studying, Transwell invasion (Matrigel) assay will be necessary.

3. Discussion section just basically repeats results. Conclusion is general.

There are some even gramma errors: "afte" line 161

Author Response

Dear Reviewer:

Thank you very much for your professional review of our article. As you pointed out, there are several problems that need to be addressed. Following your kind suggestions, we have extensively revised our previous draft. The transwell invasion (Matrigel) experiment is currently being prepared, with results available in about two weeks. Meanwhile, the manuscript had be reviewed and edited by language services of ELSEVIER.

Your comments are laid out below in italicized font and our response is given in normal font. The changes/additions to the manuscript are given in yellow text.

Q1: Cell proliferation and apoptosis assay are important but the relation to other types of cell death will be helpful.

We agree with your comments that the correlation between FAM111B expression and other types of cell death is equally important. There are many apoptosis-related proteins that are also involved in other cell death pathways. In this study, we mainly focused on the effect of FAM111B on proliferation, migration and apoptosis of BLCA cells. We will further explore the relationship between FAM111B expression and other types of cell death in subsequent studies

Q2: The results of cell movement assay are convincing, but since the relation to metastasis formation is studying, Transwell invasion (Matrigel) assay will be necessary.

The transwell invasion (Matrigel) assay is indeed necessary to metastasis study. So, we are conducting this assay. But the cells we used before were in poor condition. The new FAM111B knockdown cell models are preparing. It’s estimated that the transwell invasion (Matrigel) assay will be finished in two weeks. Please allow us to update the data once the additional experimental outcomes become obtainable.

  1. Discussion section just basically repeats results. Conclusion is general.

We have included a discussion on the existing BLCA prognostic markers (P11, Line346-351), minimized repetition of experimental findings (P12, Line365-367) and incorporated a thorough discussion on apoptosis-related proteins (P12, Line380-393).

Yours sincerely,

Ning Huang

October 16, 2023

Round 2

Reviewer 1 Report

The quality of the paper was improved and my comments were properly addressed.

The quality of English is fine.

Reviewer 2 Report

Thank you for taking on board feedback and making the requested changes. A minor comment regarding the new reference added to discussion (Ref 36). Please change 'Melanie et al' into 'Fritsch et al'. It looks like Melanie is the first name of the author, whilst the surname is Fritsch.

Reviewer 3 Report

I am satisfied with correction that authors made and when the results of Transwell assay will be included, the manuscript may be published.